# Structure-guided peptide engineering of a positive allosteric modulator targeting the outer pore of TRPV1 for long-lasting analgesia

Heng Zhang [1,2,3,7], Jia-Jia Lin[2,4,7], Ya-Kai Xie[2,4], Xiu-Zu Song[5], Jia-Yi Sun[5], Bei-Lei Zhang[5], Yun-Kun Qi [6] ✉, Zhen-Zhong Xu [2,4] ✉ & Fan Yang [1,2,3] ✉

Transient receptor potential vanilloid 1 (TRPV1) ion channel is a classic analgesic target, but antagonists of TRPV1 failed in clinical trials due to their side effects like hyperthermia. Here we rationally engineer a peptide s-RhTx as a positive allosteric modulator (PAM) of TRPV1. Patch-clamp recordings demonstrate s-RhTx selectively potentiated TRPV1 activation. s-RhTx also slows down capsaicin-induced desensitization of TRPV1 in the presence of calcium to cause more calcium influx in TRPV1-expressing cells. In addition, our thermodynamic mutant cycle analysis shows that E652 in TRPV1 outer pore specifically interacts with R12 and K22 in s-RhTx. Furthermore, we demonstrate in vivo that s-RhTx exhibits long-lasting analgesic effects in noxious heat hyperalgesia and CFA-induced chronic inflammatory pain by promoting the reversible degeneration of intra-epidermal nerve fiber (IENF) expressing TRPV1 channels in mice, while their body temperature remains unaffected. Our results suggest s-RhTx is an analgesic agent as a PAM of TRPV1.

Chronic pain not only dramatically decreases the quality of life in patients, but also imposes huge social-economical costs. For instance, more than 100 million adults suffer from chronic pain with an economic cost of about 600 billion dollars annually in the United States alone[1]. Though traditional analgesics, such as opioids and nonsteroidal anti-inflammatory drugs, have been widely used against chronic pain, they show either strong side effects or relatively low efficacy, respectively. Therefore, novel analgesics with alternative mechanisms of action are needed.

Transient receptor potential vanilloid 1 (TRPV1) ion channel is a polymodal nociceptor broadly expressed in sensory neurons involved in pain sensation[2], making this channel a target for analgesia. Since TRPV1 deletion in rodents led to significant tolerance to noxious heat hyperalgesia[3,4], antagonists of TRPV1 became the first choice to develop analgesic agents targeting TRPV1 in the past decades. Pharmacological inhibition TRPV1 indeed effectively alleviates dental, rectal, and thermal pain[5,6]. However, TRPV1 also plays critical roles in thermoregulation[7], thus its antagonists suffered failures in clinical trials for the side effects inducing hyperthermia and burn injuries in patients[8]. Agonists of TRPV1 have also been used as analgesics. For instance, resiniferatoxin induces cells death in TRPV1-expressing neurons by inducing calcium overload to treat intractable cancer

[1]Department of Biophysics, Kidney Disease Center of the First Affiliated Hospital, Zhejiang University School of Medicine, Hangzhou, Zhejiang, China. [2]Liangzhu Laboratory, Zhejiang University Medical Center, Hangzhou, Zhejiang, China. [3]Alibaba-Zhejiang University Joint Research Center of Future Digital Healthcare, Hangzhou, China. [4]Department of Neurobiology and Department of Anesthesiology of First Affiliated Hospital, Zhejiang University School of Medicine, Hangzhou, Zhejiang, China. [5]Department of Dermatology, The Third People's Hospital of Hangzhou, Zhejiang Hangzhou, China. [6]Department of Medicinal Chemistry, School of Pharmacy, Qingdao University, Qingdao, Shandong, China. [7]These authors contributed equally: Heng Zhang, Jia-Jia Lin. ✉ e-mail: qiyunkun@qdu.edu.cn; xuzz@zju.edu.cn; fanyanga@zju.edu.cn

pain[9], but the irreversibility of such analgesia limits its application[10]. Therefore, it is necessary to develop novel TRPV1 modulators for better pain managements.

Positive allosteric modulator (PAM) is defined as a ligand without direct activation of its target, but potentiates the activity of agonists of targets. PAMs of TRPV1 have shown significant analgesic effect by inducing calcium overloading in previous reports[11,12]. For instance, MRS1477 is developed as a small molecule PAM of TRPV1[13]. It enhances TRPV1 activation by capsaicin to induce analgesic effects[11,14]. We have also computationally designed peptide PAMs (*De1* and *De3*) of TRPV1 with analgesic effects against heat pain in rats[12]. However, as both *De1* and *De3* bind to the intracellular ankyrin-repeat domain (ARD) of TRPV1 to modulate the channel, TAT transmembrane peptide was fused to these PAMs so that their potential of translational development is limited. To overcome this limitation, a peptidic PAM that binds to the extracellular side of TRPV1 is desired. Therefore, we resorted to the peptide toxins of TRPV1, such as DkTx[15], BmP01[16], RhTx[17], and RhTx2[18], which all bind to the outer pore of the channel.

In this work, we rationally engineer such a peptidic PAM s-RhTx based on RhTx, a potent TRPV1 agonist we discovered from the venom of the Chinese red-headed centipede[17]. s-RhTx no longer activates TRPV1 as designed. It selectively potentiates the currents of TRPV1 evoked by capsaicin and proton in a concentration-dependent manner and slows down the desensitization process, whereas it does not change the heat activation threshold of TRPV1. Our thermodynamic mutant cycle analysis demonstrates that s-RhTx specifically interacted with E652 in the outer pore of TRPV1. We further observe that in vivo the co-application of low dose of capsaicin and s-RhTx exerted clear and long-lasting analgesic effect in naive and inflammatory-pain model mice by promoting the reversible degeneration of intra-epidermal nerve fiber (IENF) expressing TRPV1 channels. Therefore, our study demonstrates the potential of s-RhTx for pain relief.

## Results

### Rational design and purification of s-RhTx
To develop a PAM of TRPV1 targeting its extracellular structures, we rationally engineered the RhTx, which is known to bind to the outer pore of TRPV1 to activate the channel. Our previous study showed that the residues of RhTx in the C terminus are critical for its function, while the residues in the N terminus are highly flexible as revealed by NMR (PDB ID: 2MVA. Fig. 1a, residues in blue)[17]. Therefore, we hypothesized that by deleting the N terminal flexible residues, the binding configuration of RhTx will be modified so that its function will be tuned. To test our hypothesis, we deleted the first three amino acids of RhTx N terminal and kept the cysteines to ensure the formation of disulfide bonds. The shortened form of RhTx, which we named s-RhTx, was chemically synthesized and purified by reverse phase (RP)-HPLC. To ensure the correct folding and disulfide bond formation, we put the linear peptide into a 0.1 M Tris-HCl buffer (see methods) and incubated overnight. Then we used reverse phase chromatography (RPC) for purification and acetonitrile was used as mobile phase. The peak of folded s-RhTx was eluted by 31% acetonitrile with 0.1% trifluoroacetic acid in gradient elution of acetonitrile (Fig. 1c). Mass spectrometry (MS) detection showed the mass of folded s-RhTx was 2624.125, while the mass of linear form was 2628.211 (Fig. 1d), indicating the successful formation of two disulfide bonds.

### s-RhTx is a positive allosteric modulator (PAM) of TRPV1
When we perfused s-RhTx alone to TRPV1-expressing HEK293 cells but observed that 30 μM s-RhTx cannot activate TRPV1 (Fig. 1e). In contrast, we found s-RhTx was able to potentiate the effect of capsaicin (10 nM) in a concentration-dependent manner in whole-cell patch-clamp recording with a sub-micromolar EC50 of 0.89 ± 0.45 μM (Fig. 1f, g). We further tested the activity of s-RhTx in calcium imaging assay, and observed that s-RhTx significantly increased intracellular

calcium fluorescence in the presence of 10 nM capsaicin (Fig. 1h, i), while 10 nM capsaicin itself cannot induce discernable calcium influx. In contrast, the unfolded s-RhTx as a linear peptide did not potentiate TRPV1 activation in either patch-clamp recordings or calcium imaging (Supplementary Fig. 1).

TRPV1 is a polymodal receptor activated by a plethora of stimuli[2], so besides potentiation of capsaicin action, we tested the effects of s-RhTx on proton or heat activation of TRPV1. For proton-activated mode, s-RhTx was able to enhance the current of TRPV1 induced by proton (pH 6.5) with the EC50 of 1.58 ± 0.50 μM (Fig. 1j, k). On the other hand, for heat activation of TRPV1, the heat activation threshold of TRPV1 channel in the absence of PAM was 37.9 ± 1.1 °C. The agonist RhTx significantly lowered the heat activation threshold of TRPV1 to 31.0 ± 1.8 °C, which was consistent with previous report[17]. Unexpectedly, s-RhTx did not change the heat activation threshold of TRPV1 channels with the threshold of 38.6 ± 0.98 °C (Fig. 1m). Our results suggested s-RhTx actually acted as a PAM in capsaicin and proton activation of TRPV1 without affecting the heat activation threshold.

To determine the subtype selectivity of s-RhTx, we tested the effect of s-RhTx on other TRP channels involved in pain, such as TRPV2, TRPV3, and TRPA1. For TRPV2, 500 μM 2-APB elicited slight current about 7.06 ± 2.25% as normalized to current activated by 4 mM 2-APB, and s-RhTx showed little potentiation on the TRPV2 current induced by 500 μM 2-APB with the current about 9.40 ± 3.64% (mean ± S.E.M., *n* = 4, Fig. 2a). Similar to its effect on TRPV2, s-RhTx had little potentiation on TRPV3 current induced by 30 μM 2-APB, with the current of 7.08 ± 3.57% and 6.91 ± 3.33% for 30 μM 2-APB alone and the cocktail of 30 μM 2-APB and s-RhTx as normalized to current activated by 1 mM 2-APB, respectively (mean ± S.E.M., *n* = 4, Fig. 2b). Moreover, s-RhTx had little effect for the potentiation of TRPA1 current induced by 30 μM AITC as well. As shown in Fig. 2c, 30 μM AITC evoked TRPA1 current for about 41.06 ± 12.85% as normalized to current activated by 1 mM AITC. In comparison, the mixture of AITC and 10 μM s-RhTx induced TRPA1 current for about 44.38 ± 13.53% (mean ± S.E.M., *n* = 4). Therefore, our results demonstrated that s-RhTx was a selective PAM of TRPV1.

### s-RhTx increased calcium influx by slowing down the desensitization of TRPV1
TRPV1 activation by capsaicin is always accompanied by a fast desensitization process in the presence of calcium, so that TRPV1 PAMs slow down the desensitization process of TRPV1 to induce local calcium overload in nociceptive afferent nerve terminus for analgesia[11,12]. Therefore, we measured the effect of s-RhTx on the TRPV1 desensitization process in the presence of calcium in whole-cell recordings clamping the voltage both at +80 mV and −80 mV. We observed that desensitization induced by the cocktail of capsaicin and s-RhTx was remarkably slowed down compared with desensitization upon capsaicin at both +80 mV and −80 mV (Fig. 3a). At +80 mV, the desensitization time constant tau was 47.37 ± 6.14 s for capsaicin and 179.3 ± 30.37 s for the cocktail of s-RhTx and capsaicin (*n* = 9–10). At −80 mV, the desensitization time constant tau was 23.87 ± 3.97 s for capsaicin and 89.4 ± 28.21 s for the cocktail of s-RhTx and capsaicin (Fig. 3b).

Moreover, we calculated the area under the curve (AUC) of the desensitization process to estimate the amount of calcium entry to cells expressing TRPV1. We employed the normalized current (Fig. 3a) for AUC calculation. Consistent with the desensitization time constant, AUC of capsaicin-induced desensitization was 72.39 ± 6.61, and 164.7 ± 25.37 for the cocktail of s-RhTx and capsaicin at +80 mV (*n* = 9–10). And at −80 mV, AUC of capsaicin-induced desensitization was 39.25 ± 5.01, and 127.5 ± 33.66 for the cocktail of s-RhTx and capsaicin at −80 mV (Fig. 3c). These results demonstrated s-RhTx remarkably increased calcium influx though TRPV1.

We also investigate the effect of s-RhTx on cell death induced by more calcium influx via TRPV1 channel using Hoechst and propidium iodide (PI) staining assay, where Hoechst labeled all cell nuclei and PI labeled cell nuclei in dead cells. Cell death ratio in HEK293T cells expressing TRPV1 channels without adding any ligand was kept low, indicating transfection of TRPV1 has little effect on cell death. However, s-RhTx induced cell death ratio in the concentration-dependent manner in the presence of 10 nM capsaicin (Supplementary Fig. 2, $n = 4$). Our results clearly suggested that s-RhTx slowed down the desensitization process and significantly enhanced the amount of calcium influx to HEK293T cells expressing TRPV1 channel.

## E649 and E652 in TRPV1 are critical for the potentiation by s-RhTx

We hypothesized that like RhTx[17], s-RhTx also binds to the extracellular side of TRPV1 to potentiate channel activation. To test this hypothesis, we performed inside-out and outside-out recordings. In the inside-out patch configuration, TRPV1 channels had no response to even 300 μM s-RhTx in the presence of 10 nM capsaicin, while 30 μM s-RhTx elicited a remarkable current in outside-out patches (Fig. 4a). These observations demonstrated that the binding site of s-RhTx located at the extracellular side of TRPV1 rather than the intracellular side.

To probe the binding configuration of s-RhTx to TRPV1, we carried out molecular docking using Rosetta suite[19] and validated the docking models with point mutations in TRPV1. Our docking results showed R12 and K22 of s-RhTx formed hydrogen bonds with E652 of TRPV1 in two adjacent subunits and K18 of s-RhTx formed hydrogen bond with E649. Alanine scan showed E649 and E652 in the outer pore domain were critical for the activity of s-RhTx (Fig. 4d). For instance, current potentiation by s-RhTx was largely diminished in the E649A mutant. However, E649A mutant did not completely abolish the potentiation by s-RhTx (Fig. 4d, g, h), while mutations at residue E652 (E652A,

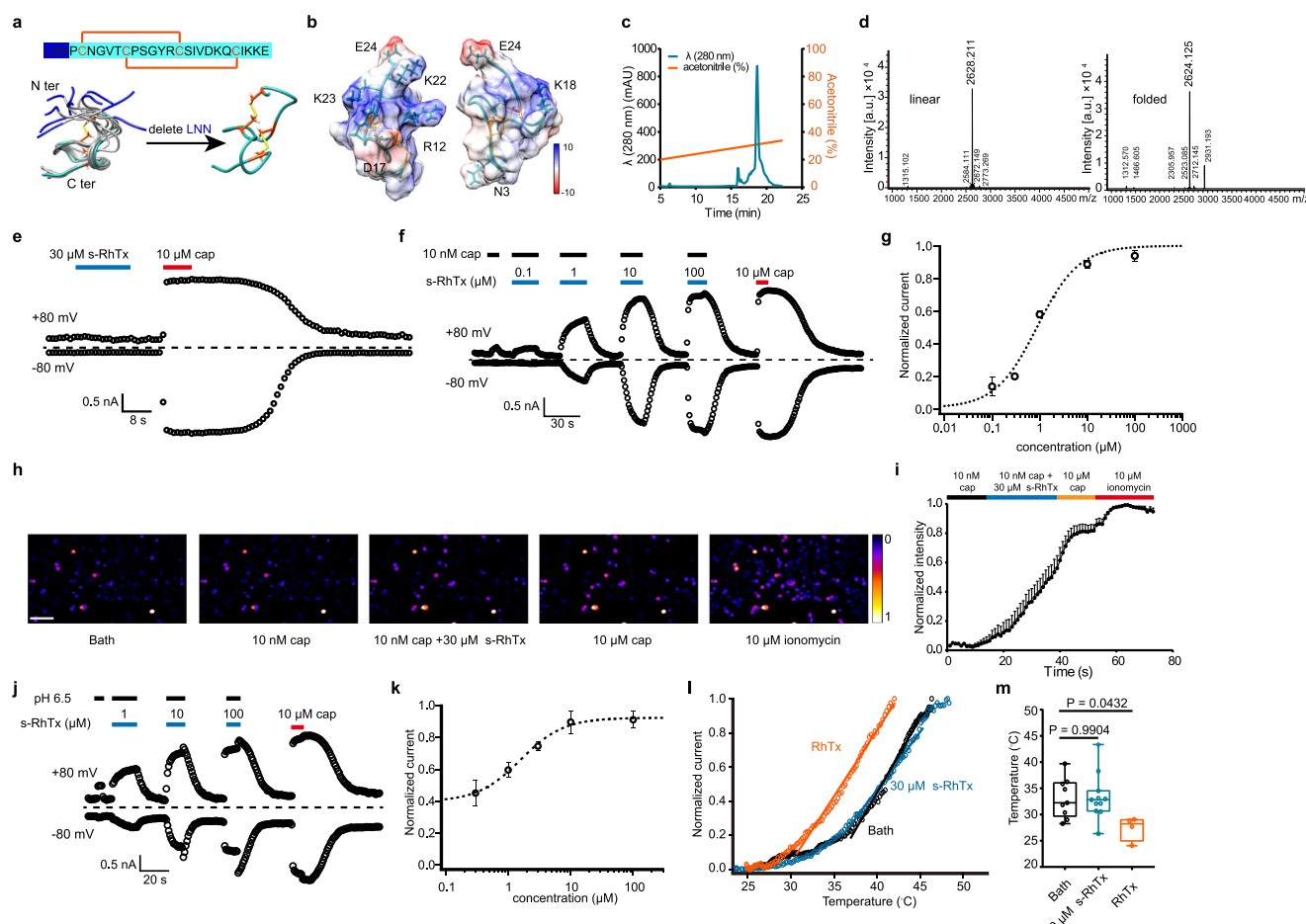

**Fig. 1 | Rational design and identification of s-RhTx as a positive allosteric modulator of TRPV1. a** Top panel, the amino acid sequence of RhTx and s-RhTx (highlighted in cyan); bottom panel, structure comparison of RhTx (PDB ID: 2MVA, colored in gray; the extra three residues in N terminus highlighted in blue) and s-RhTx (colored in cyan and disulfide bonds were labeled in orange). **b** Structural model of s-RhTx with the electrostatic potential (kcal/mol/e) distribution shown in color (red and blue for negative and positive charge, respectively). **c** Purification of s-RhTx by reverse phase chromatography with the eluted concentration of 31% for acetonitrile at the peak. **d** Mass spectrometry analysis showed the molecular weight of linear (left) and oxidative (right) s-RhTx. **e** s-RhTx cannot activate TRPV1 channel. **f** s-RhTx potentiates the effect of 10 nM capsaicin on TRPV1 in a concentration-dependent manner. **g** Concentration–response relationships of s-RhTx in the presence of 10 nM capsaicin fitted to a Hill equation (data were represented as mean ± S.E.M., $n = 4$ biologically independent cells). **h** Calcium imaging in TRPV1-expressing HEK293T cells responded to 10 nM capsaicin, the mixture of 10 nM capsaicin and 30 μM s-RhTx, 10 μM capsaicin and 10 μM ionomycin, respectively, scale bar = 100 μm. **i** The normalized fluorescence intensity of left panel (mean ± S.E.M.; $n = 6$ biologically independent cells). **j** s-RhTx concentration dependently potentiates the effect of proton on TRPV1. **k** Concentration-response relationships of s-RhTx in the presence of proton fitted to a Hill equation (mean ± S.E.M.; $n = 4$ biologically independent cells). **l** Representative heat activation of TRPV1 channels respond to RhTx or s-RhTx. **m** Box and whisker plot of heat activation thresholds of TRPV1 channels in the presence of RhTx and s-RhTx (Data in box and whisker plot were given from the minima to maxima, the bounds of box represent the 25th percentile and 75th percentile and the middle line of box is the median. Data were represented as mean ± S.E.M. from biologically independent cells; $n = 11$ for bath, $n = 10$ for s-RhTx, and $n = 4$ for RhTx; one-way ANOVA, $F (2, 21) = 3.733$, the exact $P$ values were labeled in the panel). Source data are available as a Source Data file.

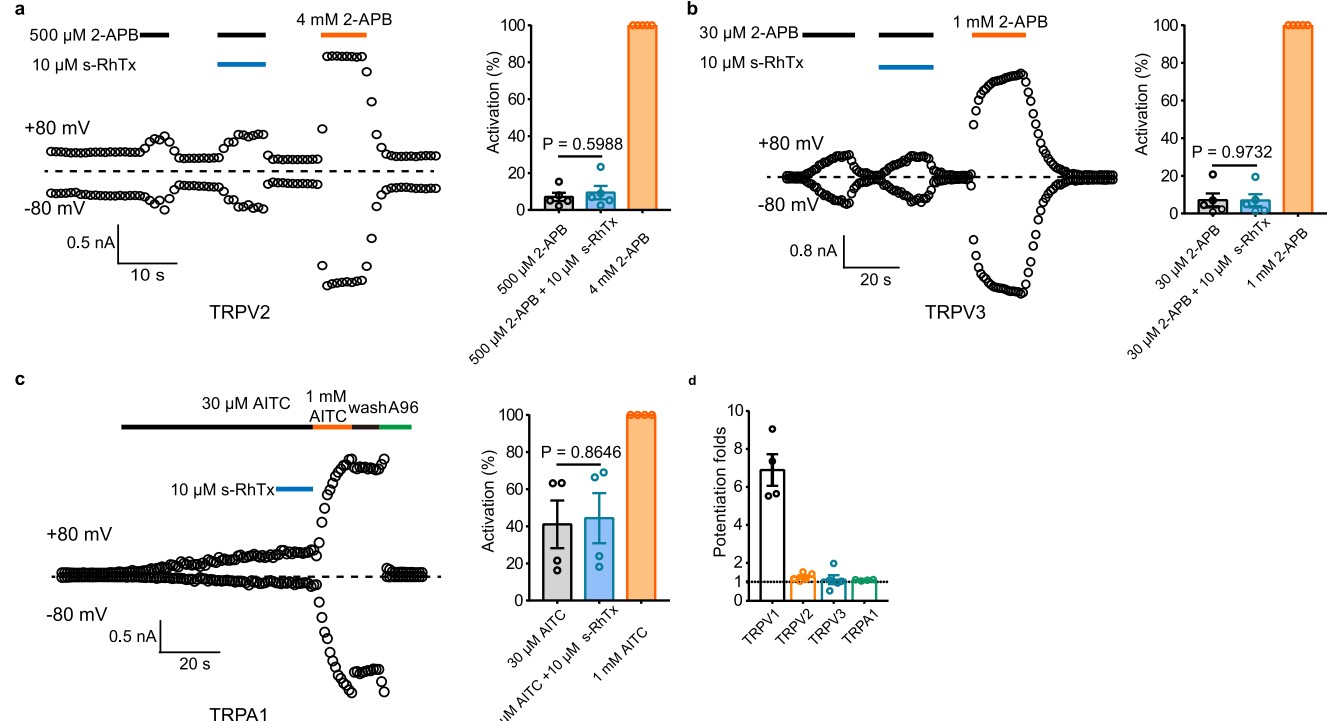

**Fig. 2 | Selectivity evaluation of s-RhTx on TRPV2, TRPV3, and TRPA1 channels expressed in HEK293T cells. a** Effect of s-RhTx on TRPV2 in the presence of 500 μM 2-APB (left) and summary of normalized current amplitude (right) (mean ± S.E.M.; *n* = 4 biologically independent cells); two-sided *t* test, *P* = 0.5988, *t* = 0.55, df = 8. **b** Effect of s-RhTx on TRPV3 in the presence of 30 μM 2-APB (left) and summary of activation (right) (mean ± S.E.M.; *n* = 4 biologically independent cells);

two-sided *t* test, *P* = 0.9732, *t* = 0.035, df = 8. **c** Effect of s-RhTx on TRPA1 in the presence of 30 μM AITC (left) and summary of activation (right) (mean ± S.E.M.; *n* = 4 biologically independent cells); two-sided t test, *P* = 0.8646, *t* = 0.18, df = 6.; A960797, a potent inhibitor of TRPA1. **d** Representative potentiation folds of s-RhTx on TRPV1, TRPV2, TRPV3, and TRPA1 channels (mean ± S.E.M.; *n* = 4 biologically independent cells). Source data are available as a Source Data file.

E652K) virtually eliminated the activity of s-RhTx on TRPV1 (Fig. 4d, e and g). Notably, E652D greatly restored the activity of s-RhTx, indicating the negative charged of residue 652 was critical for the binding and activity of s-RhTx on TRPV1 channel. Interestingly, though our previous research demonstrated that RhTx binds to four residues L461, D602, Y632, and T634 in the outer pore domain to activate TRPV1 channel[17], here we observed that L461G and T634A mutants did not attenuate the activity of s-RhTx. We reasoned that by removing the three residues in N terminus from RhTx, the s-RhTx assumed a distinct binding configuration to TRPV1 channel (Fig. 4d, e, and g).

To further verify the binding configuration of s-RhTx on TRPV1 channel, we quantified capsaicin-induced desensitization of TRPV1 mutants (Supplementary Fig. 3). s-RhTx slowed down the desensitization process of wide type (WT) TRPV1 induced by capsaicin (Fig. 3b). If E649 and E652 were the critical sites for the activity of s-RhTx, in the alanine mutants of the two residues, the desensitization time constant measured when capsaicin was applied alone or together with s-RhTx would be similar. As expected, E649A, E652A, and E652K significantly abolished the increase of desensitization time constant when the cocktail of capsaicin and s-RhTx was applied. Conversely, E652D dramatically recovered the increase of desensitization time constant in the presence of capsaicin and s-RhTx (Fig. 4f). As negative controls, other mutants such as L461G, T634A (the binding sites of RhTx), K604E, and K657E did not affect the slowing down of desensitization time constant by s-RhTx (Fig. 4f). These results were fully consistent with our alanine scan regarding the binding configuration of s-RhTx (Fig. 4e).

In addition, we also performed calcium imaging assay to further confirm the above results (Fig. 4g). Consistent with our patch-clamp recordings, on E649A mutant, s-RhTx caused 23.6 ± 0.1% enhancement

of intracellular calcium (Ca$^{2+}_i$) in response to 10 nM capsaicin (*n* = 13; Fig. 4h), which was much smaller than the Ca$^{2+}_i$ observed in WT TRPV1 (56.09 ± 8.62%, *n* = 12, Fig. 1k). Moreover, s-RhTx merely led to a 7.8 ± 0.02% rise of Ca$^{2+}_i$ on E652K mutant (*n* = 15; Fig. 4h), so that the effect of s-RhTx was largely reduced by this mutation. In contrast, s-RhTx increased the Ca$^{2+}_i$ fluorescence to 47.5 ± 0.05% on E652D mutant (*n* = 45, Fig. 4g, h). Therefore, both our electrophysiology and calcium imaging results confirmed the critical role of E649 and E652 for the potentiation effect of s-RhTx on TRPV1 channel.

**Thermodynamic mutant cycle analysis revealed the critical role of R12 and K22 in s-RhTx**

To further confirm the specific interactions between s-RhTx and TRPV1, such as the R12 and K22 of s-RhTx hydrogen bonded with E652 of TRPV1, we performed thermodynamic mutant cycle analysis. In this analysis, the ligand binding (represented by Kd) and subsequent conformational changes leading to channel opening (represented by L) were separately quantified. Kd and L were calculated by the formulae: EC50 = Kd/(1 + L) and $P_{o\,max}$ = L/(1 + L), when $P_{o\,max}$ and EC50 are experimentally measured[20,21]. Previously we have successfully performed this analysis to establish the interactions between peptide toxin BmP01 and TRPV1[16]. Four mutants including R12Q, R12K, K22Q, K22R of s-RhTx, and TRPV1 E652D mutant were tested. The chemically synthesized and folded s-RhTx mutants exhibited single-peaked elution profiles (Supplementary Fig. 4), indicating they were well folded.

We first evaluated the effect of wild-type s-RhTx on TRPV1 E652D mutant. We found s-RhTx exhibited a larger EC50 of 7.16 ± 1.42 μM, while the maximum open ability was decreased to 0.66 ± 0.06 (Fig. 5a, b, and Supplementary Fig. 5). Then we observed the effect of s-RhTx R12Q mutant on both WT TRPV1 and E652D mutant. Removal of the positively

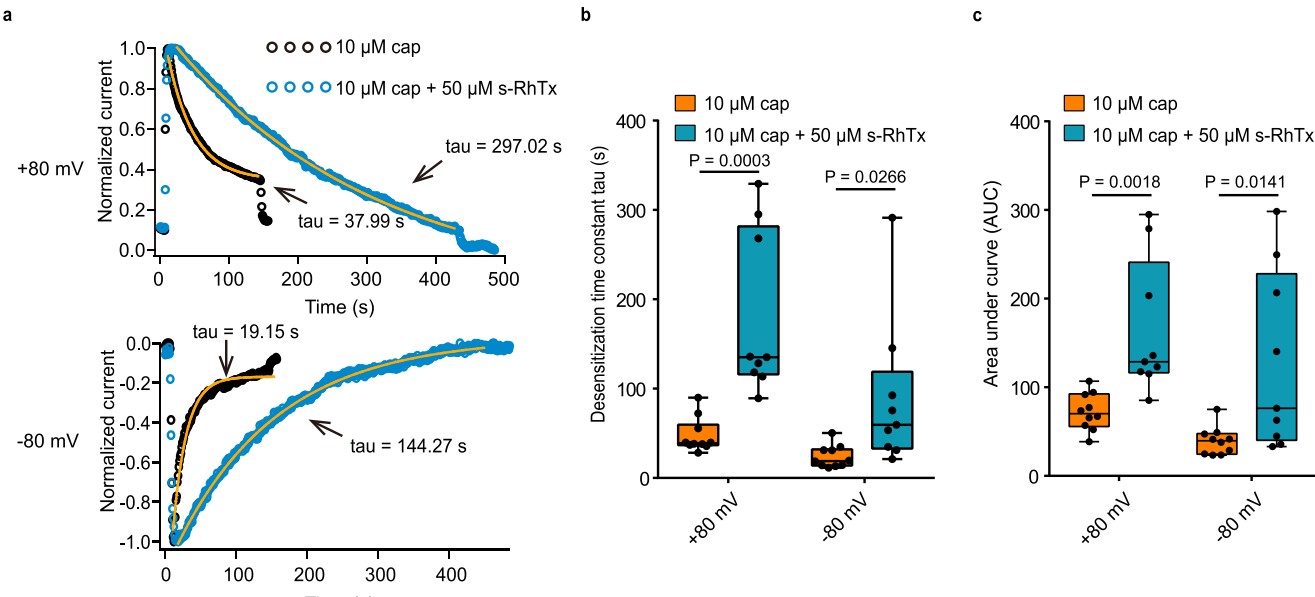

**Fig. 3 | s-RhTx slowed down the desensitization process of TRPV1 in the presence of calcium and 10 µM capsaicin. a** Comparison of the time constants tau of acute desensitization of TRPV1 in response to 10 µM capsaicin (in black) or the mixture of 10 µM capsaicin and 50 µM s-RhTx (in blue) at +80 mV (top panel) and −80 mV (bottom panel). **b** Summary of desensitization time constant of **a** (Data in box and whisker plot were given from the minima to maxima, the bounds of box represent the 25th percentile and 75th percentile and the middle line of box is the median. Data were represented as mean ± S.E.M. from biologically independent cells; *n* = 10 for cap and *n* = 9 for cap + s-RhTx; Two-sided t-test, the exact *P* values were labeled in the panels. **c** Summary of area under curve (AUC) of **a** (Data in box and whisker plot were given from the minima to maxima, the bounds of box represent the 25th percentile and 75th percentile and the middle line of box is the median. Data were represented as mean ± S.E.M. from biologically independent cells; *n* = 10 for cap and *n* = 9 for cap + s-RhTx. Two-sided *t* test, the exact *P* values were labeled in the panels). Source data are available as a Source Data file.

charged side chain by R12Q led to a significant right shift on concentration–response curve with lower open probability on both WT TRPV1 and E652D mutant, which corresponded to a much larger Kd (390.6 ± 47.12 µM) on the TRPV1 WT: R12Q pair as compared with the TRPV1 WT: s-RhTx pair (32.06 ± 16.14 µM) (Fig. 5c). Moreover, lower open probability caused a dramatically decreased *L* value: *L* value calculated for the WT TRPV1 WT: s-RhTx pair was 12.55 ± 5.34, while *L* for on the TRPV1 WT: R12Q pair was 3.37 ± 1.19 (Fig. 5d). These results clearly suggested the binding of s-RhTx R12Q mutant was weaker and it is more difficult for the s-RhTx R12Q mutant to open TRPV1 upon binding. Similar results were observed in other three s-RhTx mutants: Kd values of the three mutants became larger compared with s-RhTx on WT and E652D TRPV1. On the other hand, *L* value of the three mutants on TRPV1 and E652D became significantly smaller (Fig. 5e–m).

Previous studies including our own have shown that if the measured coupling energy is larger than 1.5 kT (or 0.89 kcal/mol at 24 °C), a specific interaction can be reliably assumed[20–23]. We further calculated the coupling energy between the channel and R12 or K22 mutants of s-RhTx based on the Kd values. As expected, R12Q, R12K, K22Q, or K22R mutants of s-RhTx showed larger coupling energy over the threshold with the E652D mutant of TRPV1 (Fig. 5n), supporting specific interactions between E652 of TRPV1 and R12 or K22 of s-RhTx predicted by our molecular docking (Fig. 5o).

### s-RhTx exerts long-lasting analgesic effects in vivo
After establishing the function of s-RhTx as a PAM of TRPV1 and its binding configuration in vitro, we then aimed to investigate the analgesic effects of s-RhTx in vivo. We reasoned that like the reported PAM of TRPV1[11], co-injection of s-RhTx with a low-dose capsaicin will transiently cause pain sensitization at the very beginning, but later such a combination should induce calcium overload to ablate the TRPV1-expressing and pain-sensing nerve terminals. Indeed, when we firstly examined capsaicin-induced spontaneous pain in mice within

10 minutes after intraplantar injection, mice injected with vehicle (saline) or folded s-RhTx did not exhibit obvious spontaneous pain behavior (Fig. 6a, b). Injection with low-dose capsaicin (200 ng in 20 µl) and the cocktail of capsaicin and linear s-RhTx (2 µg), which did not potentiate TRPV1 activation (Supplementary Fig. 1), caused a mild spontaneous pain behavior. In contrast, the co-application of capsaicin (200 ng) and folded s-RhTx (2 µg) evoked a robust nociceptive behavior featured as paw licking, lifting, and flinching in wildtype (WT) mice, which was absent in the *Trpv1*⁻/⁻ mice (Fig. 6a, b). These results suggested folded s-RhTx significantly potentiated the activation of TRPV1 channel to induce spontaneous pain within the first 10 min after injection in vivo.

To examine whether s-RhTx would exert analgesic effects in a long duration, we evaluated the paw withdrawal latency of mice in response to noxious radiant heat using Hargreaves test. As expected, mice injected with the combination of low-dose capsaicin (200 ng) and folded s-RhTx (2 µg) exhibited higher paw withdrawal latency responded to noxious heat stimulation than mice injected with capsaicin and linear s-RhTx (Fig. 6c). Importantly, such an analgesic effect lasted for a long time of over two weeks (Fig. 6c).

To elucidate the mechanism underlying such a long-lasting analgesia, we tested our hypothesis that co-application of s-RhTx and low-dose capsaicin-induced the ablation of epidermal nerve endings expressing TRPV1 channels. To test this hypothesis, we examined the distribution of *Trpv1*-positive (TRPV1⁺) nerve fibers expressing EYFP fusion protein (*Trpv1-EYFP*) in skin tissues from transgenic reporter mice, after injection of the cocktail of capsaicin and folded or linear s-RhTx (Fig. 6d). Injection of low-dose capsaicin (200 ng) and the cocktail of capsaicin (200 ng) with linear s-RhTx (2 µg) showed little effect on the number of *Trpv1*⁺ IENF, compared with the vehicle injection (Fig. 6e). In contrast, co-injection of capsaicin and folded s-RhTx (2 µg) significantly diminished the number of *Trpv1*⁺ IENF (Fig. 6e). Interestingly, IENF degeneration and thermal sensitivity were

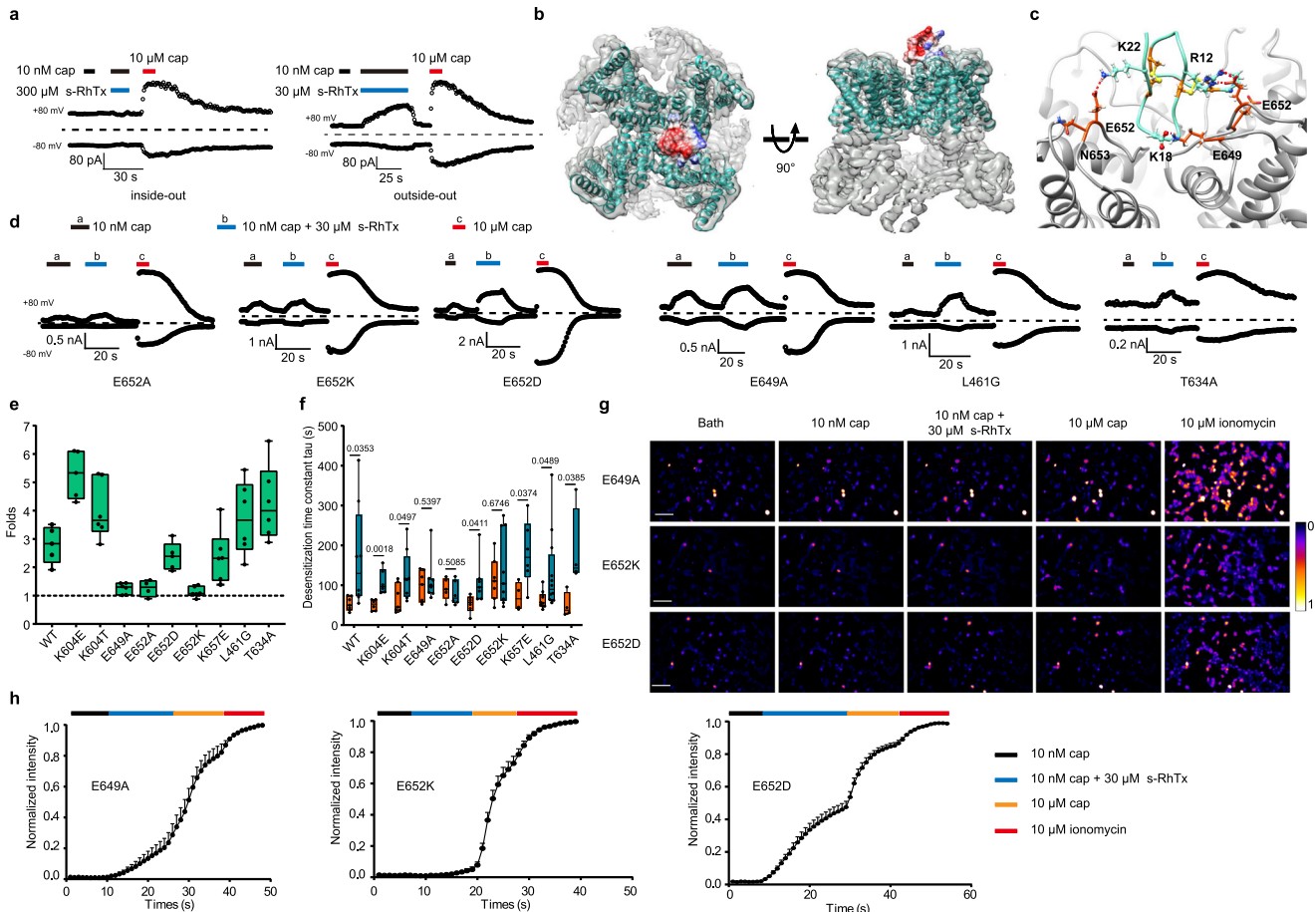

**Fig. 4 | E649 and E652 in TRPV1 at the outer pore are critical for potentiation by s-RhTx. a** Inside-out (left) and outside-out (right) recordings revealed s-RhTx binds to the extracellular domain of TRPV1. **b** Molecular docking of s-RhTx to TRPV1 (PDB ID: 3j5p). s-RhTx is colored by surface electrostatic potential (red for negatively charged and blue for positively charged). The backbone of TRPV1 is shown in cyan with the electron density map superimposed. Left panel represents the top view and the right panel representative the side view of s-RhTx binding to TRPV1. **c** Molecular docking details of s-RhTx to the outer pore of TRPV1. Hydrogen bonds are shown by green line. **d** Representative whole-cell recording of point mutants in response to 10 nM capsaicin (**a**), the mixture of 10 nM capsaicin and 10 μM s-RhTx (**b**) and 10 μM capsaicin (**c**) with the testing potential of +80 mV. **e** Representative the potentiation folds of the cocktail of s-RhTx and 10 nM capsaicin over 10 nM capsaicin for TRPV1 point mutants (Data in box and whisker plot was given from the minima to maxima, the bounds of box represent the 25th percentile and 75th percentile and the middle line of box is the median. Data were represented as mean ± S.E.M. from biologically independent cells; $n = 5$ for WT, K604E, E652D;

$n = 4$ for E652A; $n = 6$ for K604T, E649A, E652K, K657E, L461G, T634A). **f** Summary of desensitization time constant of TRPV1 point mutants (Data in box and whisker plot was given from the minima to maxima, the bounds of box represent the 25th percentile and 75th percentile and the middle line of box is the median. Data were represented as mean ± S.E.M. from biologically independent cells; For 10 μM cap, $n = 7$ for WT, K604T, E649A, E652K; $n = 4$ for K657E, T634A; $n = 5$ for E652A and $n = 6$ for K604E, E652D; For 10 μM cap + 30 μM s-RhTx, $n = 8$ for WT, K604T, E649A; $n = 4$ for T634A; $n = 6$ for K604E, E652A, K657E; $n = 7$ for E652D and $n = 12$ for L461G. Two-sided t-test, the exact $P$ values were labeled in the panels). **g** Calcium imaging in HEK293T cells expressing E649A, E652K, and E652D mutants responded to 10 nM capsaicin, the cocktail of 10 nM capsaicin and 30 μM s-RhTx, 10 μM capsaicin, and 10 μM ionomycin, respectively, scale bar = 100 μm; **h** the normalized fluorescence intensity of calcium images of TRPV1 E649A, E652K, and E652D mutants (mean ± S.E.M. from biologically independent cells; $n = 13$ for E649A, $n = 16$ for E652K and $n = 45$ for E652D). Source data are available as a Source Data file.

fully recovered at 4 weeks after the co-injection of folded s-RhTx and capsaicin (Supplementary Fig. 7a–c). Moreover, both the proportion of *Trpv1*⁺ neurons and the density of total neurons in DRGs were not significantly changed after the co-injection (Supplementary Fig. 7d, e). Therefore, the co-application of folded s-RhTx and capsaicin indeed induced reversible degeneration of TRPV1⁺ nerve endings to exert long-lasting analgesic effects in vivo.

Benefitting from capsaicin-induced nerve endings degeneration in epidermis, the local application of capsaicin has been a common analgesic treatment in chronic pain management[24]. Thus, we expected that the combined application of capsaicin and s-RhTx is even more effective for chronic pain relief. To test the effect of s-RhTx against chronic pain, we used the well-established chronic inflammatory pain model induced by Complete Freund's adjuvant (CFA) in mice[25,26] (Fig. 7a). The effects of s-RhTx against heat hyperalgesia and

mechanical allodynia in CFA-injected mice were evaluated by Hargreaves and von Frey test, respectively (Fig. 7b, c).

We observed that in the Hargreaves test, CFA-challenged mice got a slow recovery from heat hyperalgesia after the injection of capsaicin with saline or linear s-RhTx (Fig. 7b), while the injection of capsaicin with folded s-RhTx dramatically reversed heat hyperalgesia and accelerated the recovery process (Fig. 7b). Such analgesic effects of cocktail of capsaicin and s-RhTx lasted for more than three weeks (Fig. 7b). In the von Frey test, CFA mice injected with capsaicin and saline or linear s-RhTx had a robust and prolonged mechanical allodynia, which was significantly reversed by co-application of capsaicin and folded s-RhTx, suggesting that the analgesic effects of folded s-RhTx were much stronger than capsaicin alone or with the linear form of s-RhTx (Fig. 7c). Moreover, we also measured the influence of s-RhTx on body temperature of the mice. We found that the local

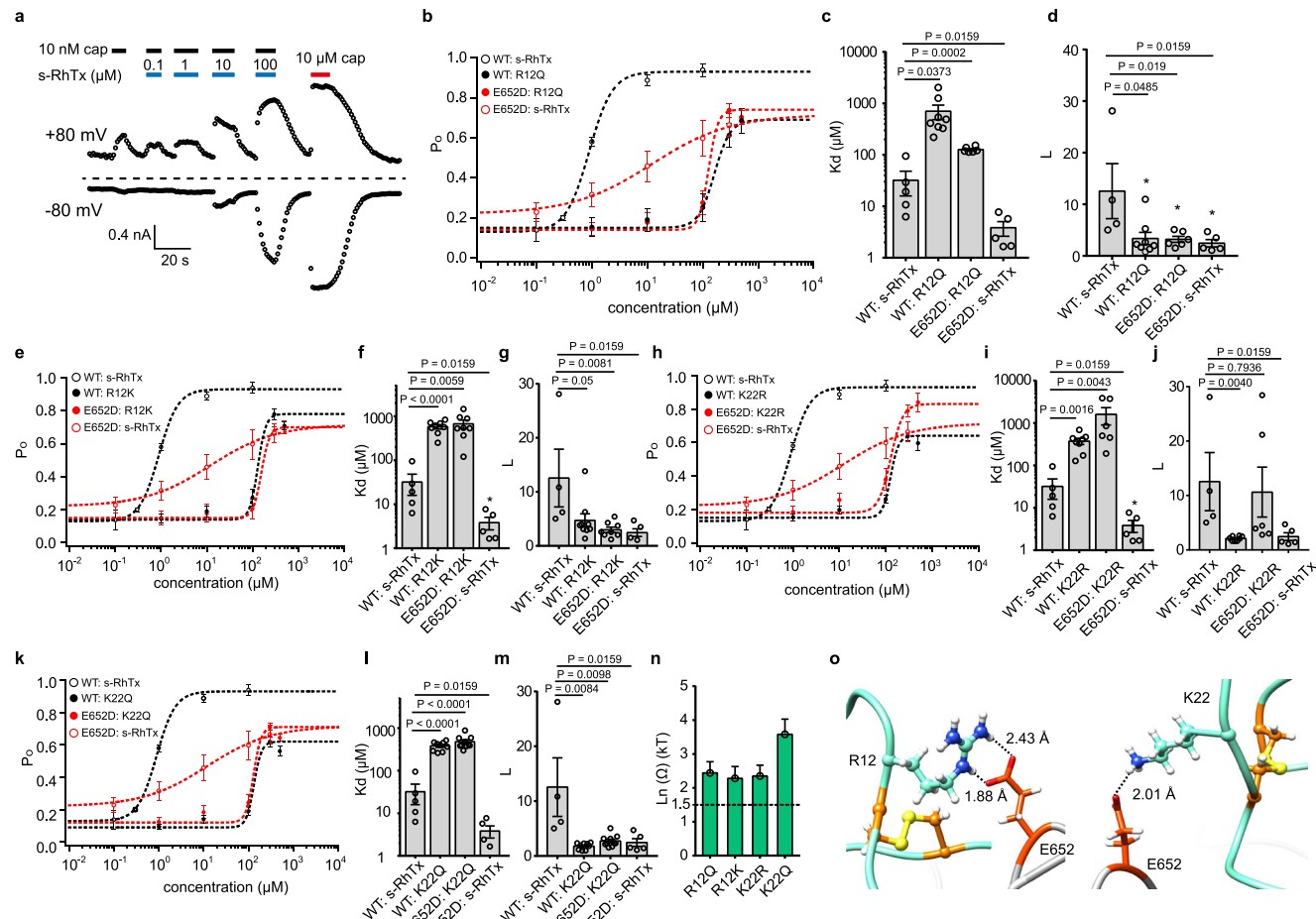

**Fig. 5 | Thermodynamics mutant cycle analysis revealed the critical role of R12 and K22 in s-RhTx. a** Concentration response of s-RhTx on E652D TRPV1 mutant. **b** Concentration–response curves of s-RhTx and R12Q on both WT and E652D TRPV1 channels (data were shown as mean ± S.E.M. from biologically independent cells. WT:s-RhTx, *n* = 5, WT:R12Q, *n* = 8, E652D:s-RhTx, *n* = 8, E652D:R12Q, *n* = 6). **c** and **d**, In the general gating of ligands, ligand binding is represented by Kd and the equilibrium constant between the closed and open states upon ligand binding is *L*. Kd and *L* values were calculated from concentration- response curves of **b** (Data were shown as mean ± S.E.M. from biologically independent cells. For Kd, *n* = 5 for WT: s-RhTx, *n* = 8 for WT: R12Q, *n* = 7 for E652D:R12Q, *n* = 5 for E652D:s-RhTx. For *L*, *n* = 4 for WT: s-RhTx, *n* = 8 for WT: R12Q, *n* = 7 for E652D:R12Q, *n* = 5 for E652D:s-RhTx. Two-sided t-test, the exact *P* values were labeled in the panels). **e** Concentration–response curves of s-RhTx and R12K on both WT and E652D TRPV1 channels (data were shown as mean ± S.E.M. from biologically independent cells. WT:s-RhTx, *n* = 5, WT:R12K, *n* = 9, E652D:s-RhTx, *n* = 8, E652D:R12K, *n* = 8). **f** and **g** Kd and *L* values calculated from concentration–response curves of **e** (Data were shown as mean ± S.E.M. from biologically independent cells; For Kd, *n* = 5 for WT: s-RhTx, *n* = 8 for WT: R12Q, *n* = 8 for E652D:R12Q, *n* = 5 for E652D:s-RhTx. For *L*, *n* = 4 for WT: s-RhTx, *n* = 9 for WT: R12Q, *n* = 8 for E652D:R12Q, *n* = 5 for E652D:s-RhTx. Two-sided t-test, the exact *P* values were labeled in the panels). **h** Concentration–response curves of s-RhTx and K22R on both WT and E652D

TRPV1 channels (data were shown as mean ± S.E.M. from biologically independent cells. WT:s-RhTx, *n* = 5, WT:K22R, *n* = 7, E652D:s-RhTx, *n* = 8, E652D:K22R, *n* = 6). **i, j** Kd and L values calculated from concentration–response curves of **h**. (Data were shown as mean ± S.E.M. from biologically independent cells; For Kd, *n*= 5 for WT: s-RhTx, *n* = 8 for WT: R12Q, *n* = 6 for E652D:R12Q, *n* = 5 for E652D:s-RhTx. For *L*, *n*= 4 for WT: s-RhTx, *n* = 8 for WT: R12Q, *n* = 6 for E652D:R12Q, *n* = 5 for E652D:s-RhTx. Two-sided t-test, the exact *P* values were labeled in the panels). **k** Concentration–response curves of s-RhTx and K22Q on both WT and E652D TRPV1 channels (data were shown as mean ± S.E.M. from biologically independent cells. WT:s-RhTx, *n* = 5, WT:K22Q, *n* = 9, E652D:s-RhTx, *n* = 8, E652D:K22Q, *n* = 10). **l, m** Kd and L values calculated from concentration–response curves of **k**. (Data were shown as mean ± S.E.M. from biologically independent cells; For Kd, *n* = 5 for WT: s-RhTx, *n* = 9 for WT: R12Q, *n* = 10 for E652D:R12Q, *n* = 5 for E652D:s-RhTx. For *L*, *n* = 4 for WT: s-RhTx, *n* = 9 for WT: R12Q, *n* = 10 for E652D:R12Q, *n* = 5 for E652D:s-RhTx. Two-sided *t* test, the exact *P* values were labeled in the panels). **n** Summary of coupling energy measurements, coupling energy value was calculated from the Kd values (data were shown as mean ± S.E.M., the exact *n* values were given in the legends **b**–**m**). **o** Representative the zoomed in view of the interaction between R12 and E652 (left panel) and K22 and E652 (right panel). Hydrogen bonds were labeled by dash lines in black. Source data are available as a Source Data file.

---

delivery of s-RhTx did not cause any changes in the body temperature of the mice (Fig. 7d). Our results suggested the cocktail of capsaicin and s-RhTx raised the mechanical and thermal thresholds in mice to relieve chronic pain without affecting body temperature (Fig. 7e).

## Discussion

In this study, we rationally designed the peptidic s-RhTx as a selective PAM of TRPV1, which exerted long-lasting analgesia in noxious heat hyperalgesia and CFA-induced inflammatory pain models in mice without changing body temperature. We further discovered that

s-RhTx as a PAM substantially increased the effect of agonists like capsaicin to cause calcium overload and degeneration of pain-sensing IENF, so that the analgesic effects lasted for two to three weeks.

Mechanistically, the binding configuration of s-RhTx to TRPV1 offered novel insights of this channel. The outer pore region of TRPV1, where s-RhTx binds to, has been known as the hot zone for channel regulation. For instance, noxious heat, proton, and divalent cations induce conformational changes in the outer pore to activate TRPV1[27–29]. However, though s-RhTx, RhTx and RhTx2 shared the similar core amino acid sequence and all bind to the outer pore

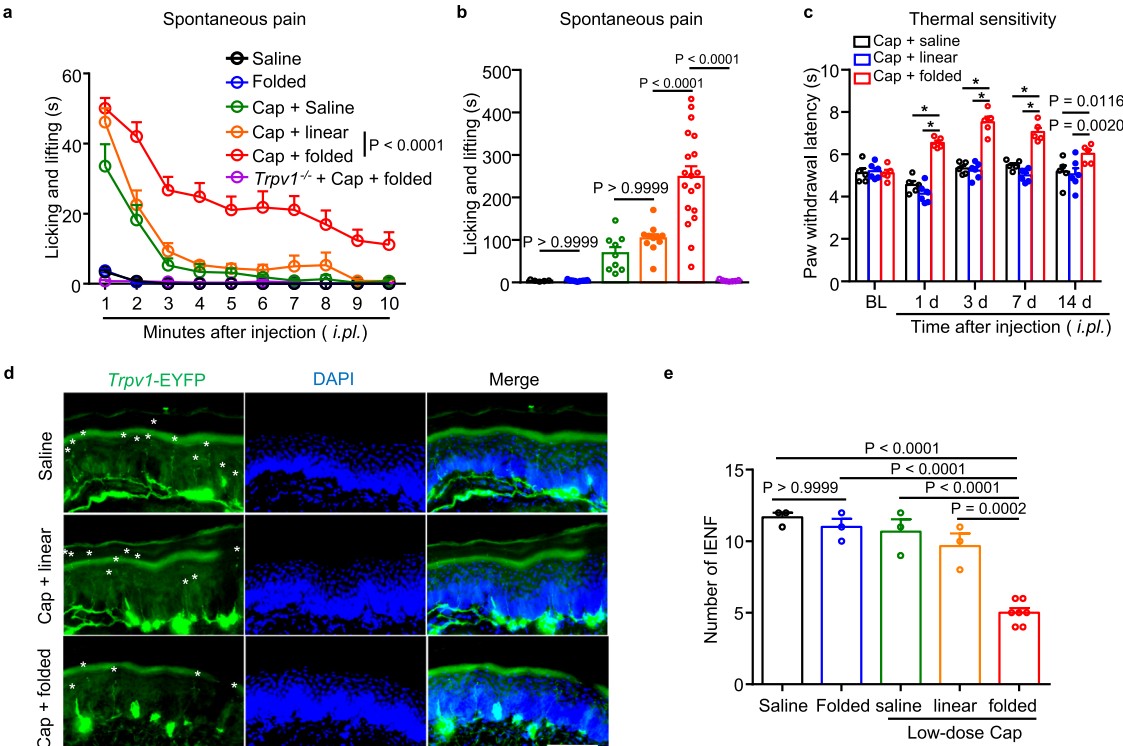

**Fig. 6 | Co-application of s-RhTx with low-dose capsaicin promotes IENF degeneration to exert long-lasting analgesic effect on thermal sensitivity in mice.** Effect of folded s-RhTx on capsaicin-induced spontaneous pain. The time spent in withdrawal, flinching, or licking the injected paw was recorded and blindly counted every minute after injection (**a**); and total time of spontaneous pain within 10 mins after injection (**b**). 2 μg of folded s-RhTx or linear s-RhTx with capsaicin (200 ng) in 20 μl saline was injected. Saline: $n = 5$, Folded: $n = 9$, Cap + saline: $n = 9$, Cap + linear: $n = 11$, Cap + folded: $n = 19$, $Trpv1^{-/-}$ + Cap + folded: $n = 7$ biologically independent mice. $P$ value was labeled in the figure, two-way ANOVA for **a**, one-way ANOVA for **b**. **c** Effect of s-RhTx on thermal sensitivity. 2 μg of folded-s-RhTx or linear-s-RhTx with capsaicin (200 ng) in 20 μl saline was injected. Data

were shown as mean ± S.E.M., Cap + saline: $n = 5$, Cap + linear: $n = 6$, Cap + folded: $n = 5$ biologically independent mice. *$P < 0.0001$ in two-way ANOVA. **d** Representative images of the immunohistochemical analysis showing $Trpv1^+$ nerve fibers degeneration in epidermis from $Trpv1$-Ai32 mice after the treatment by folded s-RhTx with low-dose capsaicin (200 ng). Scale bars, 100 μm. Every asterisk represents one fiber. Two independent experiments were conducted with similar results. **e** Quantification of $Trpv1^+$ nerve fibers in epidermis after the treatment with s-RhTx and capsaicin. Saline: $n = 3$, Folded: $n = 3$, Cap + saline: $n = 3$, Cap + linear: $n = 3$, Cap + folded: $n = 7$ biologically independent mice. P value was labeled in the figure, one-way ANOVA. All data were shown as mean ± S.E.M. Source data are available as a Source Data file.

region[17,18], they exhibited distinct mode of action. While RhTx activates TRPV1 by manipulating its heat activation machinery, s-RhTx with truncation in the N terminus cannot directly activate the channel and does not affect the heat activation. In contrast, RhTx2 with four more residues in the N terminus activates TRPV1 like RhTx, but it induced rapid current desensitization[18]. When the docking models of these three peptides were compared (Supplementary Fig. 6), we observed that despite the similar binding sites at the outer pore of TRPV1, their binding configurations were still different. Therefore, our observations here and before suggested that the outer pore of TRPV1 is highly sensitive to ligand binding, making this region attractive for developing TRPV1 modulators with distinct modes of action.

TRPV1 plays a critical role in the pathogenesis of chronic pain[30–33], and has been a long-pursued target for pain management. To circumvent the side effects of the first-generation channel inhibitors occurred in clinical trials[5], alternative strategies such as the PAMs of TRRV1 have been developed. For instance, the small molecule PAM MRS1477 and our de novo designed peptide PAM TAT-*De3* all showed significant analgesics effects in vivo[11,12]. Compared to these PAMs, our s-RhTx binds to the extracellular side of TRPV1 to potentiate the channel, so that it needs no other auxiliary component, such as the TAT transmembrane peptide in TAT-*De3*, is needed. Moreover, s-RhTx is composed of only 24 amino acids, which is much smaller than the ~100 amino acids in TAT-*De3*, so s-RhTx is less immunogenic. In fact,

s-RhTx is comparable in size as the FDA-approved analgesic peptide drug ziconotide, which is composed of 25 amino acids and engineered based on an omega-conotoxin[34,35]. Furthermore, the analgesic effects of a single injection of s-RhTx lasted for at least three weeks in vivo (Fig. 7b, c), which were substantially longer than those of MRS1477 and TAT-*De3* (about two weeks). Therefore, s-RhTx holds the potential as a promising starting point for analgesic drug developments targeting TRPV1 channel.

## Methods

### Chemicals

Capsaicin and ionomycin were purchased from MedChemExpress (MCE), 2-APB, AITC and CFA were purchased from Sigma-Aldrich.

### Animals

Adult C57 BL/6 mice (male, 8–10 weeks) and $Trpv1^{-/-}$ mice (Jax Strain #:003770) were used for behavioral studies. To specifically label $Trpv1$-positive nerve fibers in skin tissues, $Trpv1$-Cre mice[36] (Jax Strain #:017769) were crossed with floxed reporter lines (Ai32, Jax Strain #:012569 or Ai14, Jax Strain #:007914) to get the transgenic reporter ($Trpv1$-Ai32 or $Trpv1$-Ai14) mice. Mice were group-housed at 22–24 °C and relative humidity of 30–70% under the 12 h light-dark cycle with access to standard food and water *ad libitum*. All the animal procedures were approved by the Institutional Animal Care & Use Committee (IACUC) of Zhejiang University.

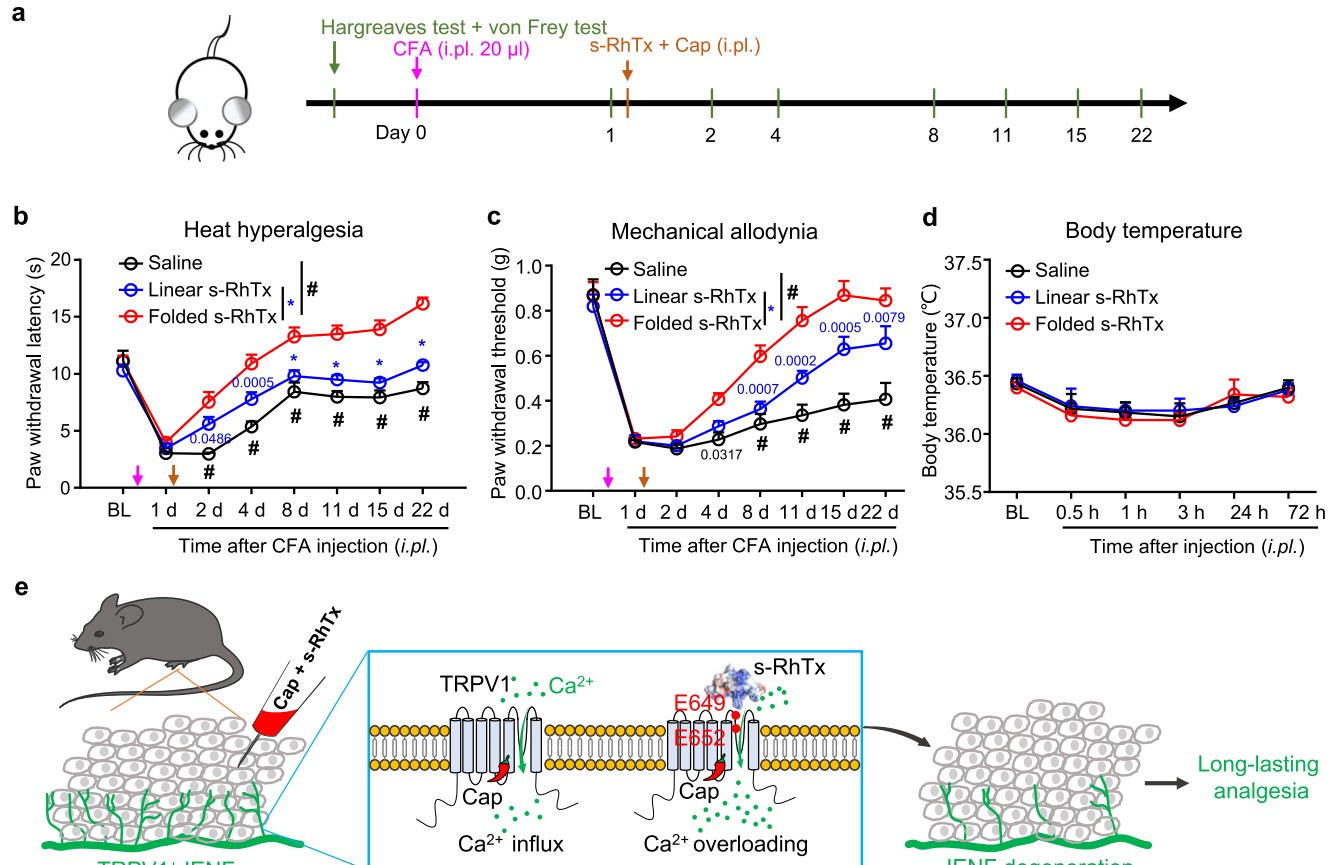

**Fig. 7 | s-RhTx relieved CFA-induced mechanical allodynia and heat hyper-algesia in chronic inflammatory pain model mice. a** Schematic illustration of the procedures to establish CFA-induced chronic inflammatory pain model and experimental time-line. 20 µl of CFA was injected first to induce a chronic pain model. 2 µg of folded s-RhTx or linear s-RhTx with capsaicin (200 ng) in 20 µl saline was injected one day after CFA-treatment. Effect of folded s-RhTx on CFA-induced heat hyperalgesia (**b**) and mechanical allodynia (**c**). BL, baseline. $n = 6$ for Saline, $n = 9$ for Linear s-RhTx, and for Folded s-RhTx, $n = 9$ biologically independent mice. * or #, $P < 0.0001$, compared with mice injected the cocktail of capsaicin and folded s-RhTx, the exact P value other than "$P < 0.0001$" was labeled in the figure, two-way

ANOVA. **d** Effect of folded s-RhTx on body temperature. Rectal temperatures in mice were measured before treatment (BL) and at 0.5, 1, 3, 24, and 72 h following folded s-RhTx or linear s-RhTx (2 µg, i.pl.) with capsaicin (200 ng) administration. $n = 6$ for Saline, $n = 5$ for Linear s-RhTx and for Folded s-RhTx, $n = 5$ biologically independent mice. Not significant in two-way ANOVA. P (saline vs folded) and $P$ (linear vs folded) are both bigger than 0.9999 at every time point. **e** Schematic diagram of the analgesic mechanism of modality-specific enhancement of TRPV1 channel opening and IENF degeneration by s-RhTx. All data were shown as mean ± S.E.M. Source data are available as a Source Data file.

## Cell culture

HEK293T cells are cultured using DMEM medium contained 10% FBS in a cell incubator with 5% $CO_2$ at 37 °C. Plasmids of murine TRPV1, TRPV2, TRPV3 and human TRPA1 are labeled with YFP or GFP as indicators for subsequent electrophysiological recordings. cDNA constructs of ion channels were transiently transfected with Lipo-fectamine 2000 (Invitrogen, Carlsbad, CA). And one or 2 days after transient transfection, electrophysiological recordings were performed.

Site mutations of mTRPV1 were generated by Fast Mutagenesis Kit (Takara Bio). Primers used to generate point mutations are summarized in Supplementary Table 1. All mutations were confirmed by sequencing.

## Toxin peptides synthesis and purification

Linear s-RhTx was synthesized from GL biochem with the purity higher than 98%. The linear s-RhTx and s-RhTx mutants (R12Q, R12K, K22Q, K22R) were dissolved in a buffer containing 0.1 M NaCl, 0.1 M Tris-HCl, 5 mM glutathione, and 0.5 mM oxidized glutathione (pH 5.6–5.8) for oxidization at 28 °C overnight. The oxidized peptide was purified by NGC system (Bio-Rad) with a reversed-phase column and detected at 280 nm. The concentrations of s-RhTx were calculated by the absorbance at 280 nm wavelength and adjusted using an extinction

coefficient 0.662 mM$^{-1}$ cm$^{-1}$ (calculated on the website https://web.expasy.org/protparam).

## Electrophysiology

Patch-clamp recordings were carried out with a HEKA EPC10 amplifier controlled by PatchMaster software (HEKA). Patch pip-ettes were prepared from borosilicate glass and fire-polished to resistance of 3–8 MΩ by P-97 puller. Both bath solution and pipette solution contained 130 mM NaCl, 10 mM glucose, 0.2 mM EDTA, and 3 mM HEPES and were adjusted to pH 7.2–7.4 with NaOH for whole-cell recordings. For solution with pH lower than 6.5, HEPES was replaced by MES. Whole-cell recordings were performed at ±80 mV. Current was sampled at 10 kHz and filtered at 2.9 kHz. All recordings were performed at room temperature (25 °C). A gravity-driven sys-tem (RSC-200, Bio-Logic) with freely rotated perfusion tubes was used for the perfusion of peptide or ligands. Bath and ligand solu-tion were delivered through separate tubes to minimize the mixing of solutions. Patch pipette holding cells was placed in front of the perfusion tube outlet for perfusion.

For desensitization of TRPV1 channel, calcium-contained solution (130 mM NaCl, 5 mM KCl, 2 mM $MgCl_2$, 2 mM $CaCl_2$, 10 mM glucose, and 10 mM HEPES, adjusted to pH 7.2–7.4 with NaOH) was used as bath solution and pipette solution.

## Calcium imaging

Calcium imaging assay was performed as previously described[12,20]. In brief, HEK293T cells expressing TRPV1 were incubated with 2 μM Fluo-4 AM (Thermo Fisher) in 2 mM calcium Ringer's solution (140 mM NaCl, 5 mM KCl, 2 mM MgCl$_2$, 2 mM CaCl$_2$, 10 mM glucose, and 10 mM HEPES, pH 7.4). Fluorescence images of HEK293T cells were acquired with Nikon Eclipse Ti2 microscope with optiMOS charge-coupled device camera controlled by the Ocular Software (Molecular Devices). Fluo-4 AM was excited at 500/20-nm excitation, and fluorescence emission was detected at 535/30-nm. Fluorescence images were analyzed with Fiji software and Office Excel.

## Cell death assay

Cell death assay was performed as previous described[37]. In brief, HEK293T cells were seeded in six-well plates and randomly divided into different groups after transfection. Then s-RhTx or capsaicin was incubated for 18 h before the determination of cell death. Hoechst 33258 (Sangon Biotech) was used to labeled all cell nuclei and propidium iodide (PI) (Sangon Biotech) was used to labeled nuclei of dead cells. In each group, five images were randomly taken under a microscope (Eclipse Ti2; Nikon, Tokyo, Japan). Cell death ratio was represented by the average of the percentage of dead cells (PI-labeled cells over Hoechst-labeled cells) from five images. Fiji software was used to count cells.

## Molecular docking of s-RhTx and TRPV1

Rosetta program suite was used to perform molecular docking of s-RhTx and Monte Carlo algorithm was employed for sampling[38,39]. The structure of s-RhTx was generated based on the structure of RhTx (PDB ID: 2MVA) and relaxed in Rosetta version 2019. The s-RhTx was put at the out pore of rTRPV1 (3j5p, relaxed in advance) for docking. 30,000 models were generated and the top 10 with lowest-energy models of them were chosen for further analysis.

## Data analysis

Data from whole-cell recordings were analyzed in Igor Pro version 5.05 (WaveMatrix). Hill Equation was used to fit the concentration-response curves for the calculation of EC50. Desensitization constant tau was obtained by fitting exponential function to the TRPV1 desensitization process.

To perform thermodynamic cycle analysis, Kd values of four channel-ligand combinations (WT channel, s-RhTx: Kd_1; Mutant channel, s-RhTx: Kd_2; WT channel, mutant s-RhTx: Kd_3; Mutant channel, mutant s-RhTx: Kd_4) were determined separately. The strength of coupling was determined by the coupling energy (kT multiplied by LnΩ, where $k$ is the Boltzmann constant and $T$ is temperature in Kelvin). LnΩ was calculated by the following equation:

$$\text{Ln}\,\Omega = \text{Ln}\left(\frac{\text{Kd}\_1 \cdot \text{Kd}\_4}{\text{Kd}\_2 \cdot \text{Kd}\_3}\right)$$

## Drug injection

Linear or folded s-RhTx was dissolved and diluted with saline. Peptide (2 μg, 20 μl) supplied with capsaicin (200 ng, Sigma-Aldrich) was delivered into the mice hindpaw via intraplantar injection (i.pl.). 20 μl saline or capsaicin (200 ng) without peptides was injected as negative control. Complete Freund's adjuvant (CFA) was injected at a volume of 20 μl to induce chronic inflammatory pain.

## Behavioral tests in mice

Spontaneous pain was measured by counting the time (s) mice spent on licking, lifting, or flinching the affected hind paws over a 10-min period after i.pl. injection.

Mechanical allodynia was assessed via von Frey test as previously reported[25]. Mice were placed in a plexiglass chamber upon the elevated metal mesh with 1 cm openings and allowed to acclimatize for 1 h. To determine mechanical thresholds of paw withdrawal, a series of von Frey filaments (0.02–2.56 g, Stoelting Co.) with increasing stiffness were used according to Dixon's up-down method.

Thermal sensitivity was tested using a Hargreaves radiant heat apparatus (IITC Life Science) as previously reported[40,41]. For heat hyperalgesia test in CFA model, the basal paw withdrawal latency was adjusted to 9 to 12 s, with a cutoff of 20 s to prevent tissue damage. For thermal sensitivity test, the basal paw withdrawal latency was adjusted to 4 to 6 s, with a cutoff of 10 s to prevent tissue damage. Withdrawal latency were averaged from three consecutive tests with a 5 min interval between measurements.

## Immunohistochemistry and quantification of intra-epidermal nerve fiber (IENF) density

After appropriate survival times, the mice were deeply anesthetized with isoflurane and perfused through the ascending aorta with PBS, followed by 4% PFA. After the perfusion, the hind paw glabrous skins and L4-L6 DRGs were removed and postfixed in the same fixative overnight. Skin sections (30 μm) and DRG sections (10 μm) were cut in a cryostat. Sections were blocked with 2% BSA for 2 h at room temperature and then incubated with Deep-red 640/660-conjugated Nissl (1:1000) at room temperature for 2 h. Sections were then incubated with DAPI (1:10,000, Vector Laboratories, catalog H-1200) for 5 min, followed by three times washing with PBS for 5 min each. The stained and mounted sections were examined with a Nikon fluorescence microscope, and images were captured with a charge-coupled device Spot camera. For quantification analysis, at least three sections from each mouse were selected, and at least three mice were analyzed in each group. To evaluate *Trpv1*-positive nerve fiber degeneration, transgenic reporter (*Trpv1*-Ai32 or *Trpv1*-Ai14) mice were used for the quantification of intra-epidermal nerve fiber (IENF) density. IENF density was calculated by counting the number of IENF crossing the epidermis-dermis boundary within unit length.

## Statistics and reproducibility

All statistical data are represented as mean ± S.E.M. The two-sided *t-test* was used between two groups to analyze the statistical significance. For more than two groups, one-way ANOVA was used; when comparing the effect of two factors on multiple groups, two-way ANOVA with a Bonferroni post-test was used. Statistical analyses were performed using GraphPad Prism 7.0 software (GraphPad Inc., La Jolla, CA). N.S. indicates not significant. *, **, ***, and **** indicate $P < 0.05$, $P < 0.01$, $P < 0.001$ and $P < 0.0001$, respectively.

For the representative experiments, experiments were performed over three separate times with similar results and the representative data were reproducible.

## Reporting summary

Further information on research design is available in the Nature Portfolio Reporting Summary linked to this article.

## Data availability

The data that support this study are available from the corresponding authors upon reasonable request. The PDB files used in this study are 2MVA [https://doi.org/10.2210/pdb2MVA/pdb] and 3J5P [https://doi.org/10.2210/pdb3J5P/pdb]. The data underlying Figs. 1–7; and Supplementary Figs. 1, 2 and 5 are provided as a Source Data File. Source data are provided with this paper.

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

## Acknowledgements

We are grateful to our lab members for their assistance and discussion. We thank Dr. Cheng Ma from the Core Facilities, Zhejiang University School of Medicine for the technical support. This work was supported by funding from National Natural Science Foundation of China (32122040 and 31971040 to F.Y.; 81971050 and 82171206 to Z.Z.X.; 22177058 to Y.K.Q.); Zhejiang Provincial Natural Science Foundation of China (LR20C050002 to F.Y. and LZ18C090002 to

Z.Z.X.); National Key R&D Program of China (2021ZD0202703 to Z.Z.X.). This work was supported by Alibaba Cloud and the MOE Frontier Science Center for Brain Science & Brain-Machine Integration, Zhejiang University.

## Author contributions

F.Y., Z.Z.X., H.Z., and Y.K.Q. conceived the study. H.Z. conducted the experiments including peptide folding, patch-clamp recordings, and molecular docking; H.Z, J.Y.S., and B.L.Z. conducted patch clamp recordings. J.J.L. and Y.K.X. conducted the animal behavior and immunohistochemistry experiments; Y.K.Q. synthesized the peptides; H.Z., J.J.L., Z.Z.X., and F.Y. participated in data analysis and manuscript writing. F.Y., Z.Z.X., Y.K.Q., and X.Z.S. supervised the project and prepared the manuscript.

## Competing interests

The authors declare no competing interests.
