## [Peer Review File · Nature Communications]

Structure-guided peptide engineering of a positive allosteric modulator targeting the outer pore of TRPV1 for long-lasting analgesiaReviewers' Comments:

Reviewer #1:

Remarks to the Author:

This study by Zhang et al describes the rational design of a peptide, sRhTx, that binds to the outer pore of the TRPV1 channel where it acts as a positive allosteric modulator. The authors show that sRhTx has no orthosteric activity at TRPV1, but very good allosteric activity, with selective potentiation of capsaicin induced currents through sustained calcium influx. They also report specificity for TRPV1 over other TRP channels and no change in heat activation of TRPV1 in the presence of the peptide. In a rodent model of inflammatory pain, the peptide gives very long lasting analgesia when administered with low dose capsaicin. I found this to really interesting study with strong design and excellent presentation. I believe will be of interest to a wide group of readers.

I have no major changes to suggest, but I do have minor comments that may improve the manuscript:

Figure 1d. The intensity of ionomycin hasn't plateaued in this figure – are there further data points to complete this time plot?

Figure 4d and 5a. It is interesting to see these mutants don't show inward currents at hyperpolarizing potentials in the presence of low peptide and capsaicin. Is this a property of the channel (due to the mutation), or variability in the experiment?

Figure 6 and page 13 results: It is important to know how long the spontaneous pain induced by injection of capsaicin and folded peptide lasts for. If not in the Fig 6 graph, a sentence describing this in the text would be helpful.

The mechanism underlying long lasting analgesia is reported to be due to calcium influx induced degeneration of nerve fibres (Fig 6 and Fig S2). How does this differ to resiniferatoxin, which in the introduction, is described as inducing cells death in TRPV1-expressing neurons by inducing calcium overload, with limited action because of irreversibility. It is not clear how cell death can be reversible with sRhTx – please include an explanation here. If data is available, it would be interesting to see a comparison between resiniferatoxin with sRhTx in the cell death assay in Fig S2.

Figure S2. The Hoechst and PI figure is very dark – it would help to increase brightness so images can be seen

Reviewer #2:

Remarks to the Author:

This manuscript, authored by Zhang et al, reported a rationally designed peptide (S-RhTx) that serves as a positive allosteric modulator of the TRPV1 channel. The authors further demonstrated that this peptide can potentiate the function of capsaicin, leading to degeneration of intra-epidermal nerve fiber (IENF) to result in long-lasting analgesic effects. This paper is clearly written and easy to read. The methodology is sound. Results are interesting and most of the data is of high quality. However, since this peptide is suggested to have its analgesic effects by degenerating IENF, the significance of this study is not extremely clear to me. In particular, I have the following comments.

1 The author stated in the Introduction section that resiniferatoxin (RTX) induces cell death to treat pain, and "the irreversibility of such analgesic limits its application". It seems the new peptide s-RhTx reported in this study has its analgesic effects via a similar mechanism. Then how RhTx might be more useful or improved over RTX?

2 s-RhTx and capsaicin were administered with intraplantar injection into the mice for testing, and

then the author found reduced IENF. However, it is not known whether the death of the neuron cell body may occur via this intraplantar injection. Since it is suggested that RTX results in cell death, the effect of s-RhTx on the cell body in mice should be examined.

3 The author reported that the analgesic effects can last for 2-3 weeks. Since the mechanism of analgesic effects seem to come from degeneration of IENF, would the analgesic effects last even longer? Did the author test the mice several months later after the application of s-RhTx and capsaicin? would the IENF grow back eventually from the same neuron or be compensated by different neurons?

4 In Figure 6e, folded s-RhTx only without capsaicin should be added as a control to rule out the possibility that s-RhTx itself is toxic and leads to the degeneration of IENF.

5 Would the degeneration of IENF mediated by s-RhTx and capsaicin lead to local neuroinflammation or recruitment of immune cells? Also, there are several ways of degeneration of IENF (e.g., "dying back" of the axon). What is the type of IENF degeneration in this study?

6 local injections of s-RhTx are not likely to change body temperature. If the authors wish to discuss the effect of s-RhTx/capsaicin on body temperature (or other potential side effects), a different experiment should be performed.

Reviewer #3:

None

The following is a point-to-point response to Reviewers' comments
(Reviewers' comments are in italics)

Reviewer #1 (*Remarks to the Author*):

This study by Zhang et al describes the rational design of a peptide, sRhTx, that binds to the outer pore of the TRPV1 channel where it acts as a positive allosteric modulator. The authors show that sRhTx has no orthosteric activity at TRPV1, but very good allosteric activity, with selective potentiation of capsaicin induced currents through sustained calcium influx. They also report specificity for TRPV1 over other TRP channels and no change in heat activation of TRPV1 in the presence of the peptide. In a rodent model of inflammatory pain, the peptide gives very long lasting analgesia when administered with low dose capsaicin. I found this to really interesting study with strong design and excellent presentation. I believe will be of interest to a wide group of readers.

Response: We are very thankful for the support and suggestions from Reviewer #1.

I have no major changes to suggest, but I do have minor comments that may improve the manuscript:

Figure 1d. The intensity of ionomycin hasn't plateaued in this figure – are there further data points to complete this time plot?

Response: We have performed additional calcium imaging experiments, so that the intensity of ionomycin has plateaued as shown below. We have replaced the original panel in Fig. 1d with the following panel in the revised manuscript.

Figure 4d and 5a. It is interesting to see these mutants don't show inward currents at hyperpolarizing potentials in the presence of low peptide and capsaicin. Is this a property of the channel (due to the mutation), or variability in the experiment?

Response: TRPV1 channel is a voltage-gated ion channel, so its open probability is increased at hyperpolarized membrane potential with V_{half} being larger than +100 mV (Yang *et al.*, *Advanced Science*, 2020). Therefore, TRPV1 current shows obvious outward rectification, so that when the membrane potential is clamped to +80 or -80 mV, much larger current is observed

at +80 mV than -80 mV. Co-perfusion of s-RhTx peptide and capsaicin activated TRPV1 channel, but the voltage gating of this channel remained so that the current at -80 mV was much smaller.

Figure 6 and page 13 results: It is important to know how long the spontaneous pain induced by injection of capsaicin and folded peptide lasts for. If not in the Fig 6 graph, a sentence describing this in the text would be helpful.

Response: We agree with the reviewer that it is an important question. We conducted additional experiment to test how long the spontaneous pain induced by injection of capsaicin and folded peptide lasts for. As shown below, co-application of capsaicin (200 ng) and folded s-RhTx (2 µg) induced robust spontaneous pain within the first 10 minutes (**Figure R1A&B**); however, there was no significant spontaneous pain after 10 minutes (**Figure R1B**). Thus, these new results consolidated our original conclusion that folded s-RhTx significantly potentiated the activation of TRPV1 channel to induce spontaneous pain within the first 10 minutes after injection.

Figure R1. Co-application of capsaicin and folded s-RhTx induced robust spontaneous pain in mice.

(A) Time course of spontaneous pain induced by injection of capsaicin and folded s-RhTx. The time spent on lifting, flinching, or licking the injected paw was recorded and blindly counted every minute after injection. 2 µg folded s-RhTx with or without capsaicin (200 ng) in 20 µl saline was intraplantarly injected. n = 9 mice in each group. *P < 0.05, two-way ANOVA followed by Bonferroni's post hoc test. Note that administration of folded s-RhTx alone didn't induce a spontaneous pain response in mice. (B) Total time of spontaneous pain within 10 minutes and from 10 to 15 minutes (mins) after injection. n = 9 mice in each group. N.S., not significant, *P < 0.05, two-way ANOVA followed by Bonferroni's post hoc test. All data were shown as mean ± s.e.m.

The mechanism underlying long lasting analgesia is reported to be due to calcium influx induced degeneration of nerve fibres (Fig 6 and Fig S2). How does this differ to resiniferatoxin, which in the introduction, is described as inducing cells death in TRPV1-expressing neurons by inducing calcium overload, with limited action because of irreversibility. It is not clear how cell death can be reversible with sRhTx – please include an explanation here. If data is available, it would be interesting to see a comparison between resiniferatoxin with sRhTx in the cell death assay in Fig S2.

Response: We thank the reviewer for the critical comments. We performed a few *in vivo* experiments to examine whether co-application of folded s-RhTx with low-dose capsaicin could induce neuronal cell death in mice. The results are presented in Figure S7 in the revised manuscript. Co-application of folded s-RhTx with capsaicin induced a robust IENF degeneration at 1 week, which was then fully recovered at 4 weeks after the injection (Fig. S7B-C in the revised manuscript). Furthermore, both the proportion of *Trpv1*⁺ neurons and the density of total neurons (Nissl⁺) in DRGs did not change at 1 week (robust IENF degeneration) or 4 weeks after the injection (Fig. S7D-E in the revised manuscript). These data suggest that co-application of s-RhTx with capsaicin did not induce the neuronal cell death in mice *in vivo*.

Unfortunately, resiniferatoxin is unavailable to purchase in China or United States now, which prevented a comparison between resiniferatoxin with s-RhTx in the cell death assay in Fig S2.

Figure S7. Co-application of s-RhTx with low-dose capsaicin induced the reversible IENF degeneration but not the loss of DRG neurons. (A) Thermal sensitivity of mice with the intraplantar injection of folded s-RhTx (2 μ g) with or without capsaicin (200 ng). $n = 6-9$ mice in each group. * or #, $P < 0.05$, two-way ANOVA followed by Bonferroni's post hoc test. (B) Representative images showing *Trpv1*⁺ nerve fibers degeneration in epidermis from *Trpv1*-Ai14 mice after the treatment by folded s-RhTx with capsaicin. Scale bars, 100 μ m. Every asterisk represents one fiber. (C) Quantification of *Trpv1*⁺ IENF in epidermis at 1 week (left) or 4 weeks (Right) after the treatment with s-RhTx and capsaicin. N.S., not significant, *, $P < 0.05$ in unpaired student's t test, $n = 3$ mice per group. (D) Representative immunohistochemical images showing *Trpv1*⁺ and Nissl⁺ neurons in DRGs from *Trpv1*-Ai14 mice after the treatment by folded s-RhTx with or without capsaicin. (E) Quantification of *Trpv1*⁺ and Nissl⁺ neurons in DRGs at 1 week (left) or 4 weeks (Right) after the treatment by folded s-RhTx and capsaicin. N.S., not significant in unpaired student's t test, $n = 3$ mice per group. Scale bars, 100 μ m. All data were shown as mean \pm s.e.m.

Figure S2. The Hoechst and PI figure is very dark – it would help to increase brightness so images can be seen.

Response: We appreciate the suggestion of reviewer #1. We have increased the brightness in the revised manuscript as shown below. The original figure has also been replaced.

Reviewer #2 (Remarks to the Author):

This manuscript, authored by Zhang et al, reported a rationally designed peptide (S-RhTx) that serves as a positive allosteric modulator of the TRPV1 channel. The authors further demonstrated that this peptide can potentiate the function of capsaicin, leading to degeneration of intra-epidermal nerve fiber (IENF) to result in long-lasting analgesic effects. This paper is clearly written and easy to read. The methodology is sound. Results are interesting and most of the data is of high quality. However, since this peptide is suggested to have its analgesic effects by degenerating IENF, the significance of this study is not extremely clear to me. In particular, I have the following comments.

Response: We thank the reviewer for the constructive suggestions, which are very helpful to improve our manuscript. To address the reviewer's concerns, we have performed additional experiments to consolidate our conclusions and highlight the significance of this study. These new results are presented as **Fig. 6A, B, E and Fig. S7** in the revised manuscript. We also provided **Response Figures R2-R4** in the point-to-point response below.

1. The author stated in the Introduction section that resiniferatoxin (RTX) induces cell death to treat pain, and “the irreversibility of such analgesic limits its application”. It seems the new peptide s-RhTx reported in this study has its analgesic effects via a similar mechanism. Then how RhTx might be more useful or improved over RTX?

Response: We thank the reviewer for pointing out this important issue. The key difference between our s-RhTx and RTX is the reversibility. RTX treatment, as reported in literature, irreversibly induces cell death. In revision of this manuscript, we conducted a series of experiments to examine whether intraplantar co-application of folded s-RhTx with low-dose capsaicin would, like RTX, induce irreversible IENF degeneration and neuronal cell death in mice *in vivo*. The results are presented in Figure S7 in the revised manuscript. Co-application of s-RhTx with capsaicin induced a robust IENF degeneration (Fig. S7B-C in the revised manuscript) and analgesic effect on thermal stimulation at 1 week after the injection (Fig. S7A). Interestingly, thermal sensitivity and IENF degeneration were fully recovered at 4 weeks after the injection of folded s-RhTx with capsaicin (Fig. S7A&C). Moreover, both the proportion

of *Trpv1*⁺ neurons and the density of total neurons (Nissl⁺) in DRGs did not change at 1 week (robust IENF degeneration) or 4 week (recovered IENF) after the injection of folded s-RhTx with capsaicin (Fig. S7D-E in the revised manuscript). These data suggest that co-application of s-RhTx with low-dose capsaicin induced the reversible IENF degeneration but not neuronal cell death in mice. Thus, our new peptide folded s-RhTx reported in this study could have fewer side effects, which is advantageous over RTX.

Fig. S7 Co-application of s-RhTx with low-dose capsaicin induced the reversible IENF degeneration but not the loss of DRG neurons. (A) Thermal sensitivity of mice with the intraplantar injection of folded s-RhTx (2 μ g) with or without capsaicin (200 ng). $n = 6-9$ mice in each group. * or #, $P < 0.05$, two-way ANOVA followed by Bonferroni's post hoc test. (B) Representative images showing *Trpv1*⁺ nerve fibers degeneration in epidermis from *Trpv1*-Ai14 mice after the treatment by folded s-RhTx with capsaicin. Scale bars, 100 μ m. Every asterisk represents one fiber. (C) Quantification of *Trpv1*⁺ IENF in epidermis at 1 week (left) or 4 week (Right) after the treatment with s-RhTx and capsaicin. N.S., not significant, *, $P < 0.05$ in unpaired student's t test, $n = 3$ mice per group. (D) Representative immunohistochemical images showing *Trpv1*⁺ and Nissl⁺ neurons in DRGs from *Trpv1*-Ai14 mice after the treatment by folded s-RhTx with or without capsaicin. (E) Quantification of *Trpv1*⁺ and Nissl⁺ neurons in DRGs at 1 week (left) or 4 week (Right) after the treatment by folded s-RhTx and capsaicin. N.S., not significant in unpaired student's t test, $n = 3$ mice per group. Scale bars, 100 μ m. All data were shown as mean \pm s.e.m.

2. *s-RhTx* and capsaicin were administered with intraplantar injection into the mice for testing, and then the author found reduced IENF. However, it is not known whether the death of the neuron cell body may occur via this intraplantar injection. Since it is suggested that RTX results in cell death, the effect of *s-RhTx* on the cell body in mice should be examined.

Response: We thank the reviewer for the constructive suggestions. We performed additional experiments to test whether there's the death of DRG neuronal cell body induced by intraplantar injection of folded s-RhTx and capsaicin. As shown in Fig. S7, the proportion of *Trpv1*⁺ neurons in DRGs did not change at both 1 week (robust IENF degeneration) and 4 week

(recovered IENF) after the injection of folded s-RhTx with capsaicin (Fig. S7D-E in the revised manuscript). Furthermore, the density of total neurons (Nissl⁺) in DRGs also did not alter at both 1 week (robust IENF degeneration) and 4 week (recovered IENF) after the injection of folded s-RhTx with capsaicin (Fig. S7D-E in revised manuscript). These data indicate that co-application of s-RhTx with low-dose capsaicin induced IENF degeneration but not the death of neuronal cell body in mice.

3 The author reported that the analgesic effects can last for 2-3 weeks. Since the mechanism of analgesic effects seem to come from degeneration of IENF, would the analgesic effects last even longer? Did the author test the mice several months later after the application of s-RhTx and capsaicin? would the IENF grow back eventually from the same neuron or be compensated by different neurons?

Response: We thank the reviewer for the critical comments. As suggested, we tested how long the analgesic effects could last after the application of s-RhTx and capsaicin in CFA-induced chronic inflammatory pain model mice. As shown in Figure R2, co-application of s-RhTx with capsaicin significantly relieved mechanical allodynia and heat hyperalgesia for 4 weeks (Figure R2 A-C) in CFA model mice.

In addition, since no DRG neuronal death was induced (Fig. S7D-E in the revised manuscript) and IENF degeneration was fully recovered at later stage after the injection of folded s-RhTx with capsaicin (Fig. S7C), it seems that the IENF could grow back likely from the same DRG neuron. However, whether the regenerated IENF were from the same neuron or different neurons need further confirmation by in vivo IENF imaging with two-photon microscopy in the future study.

Figure R2. Co-application of s-RhTx with capsaicin relieved mechanical allodynia and heat hyperalgesia in CFA-induced chronic inflammatory pain model mice. (A) Schematic illustration showing the procedure for CFA model establishment, drug treatment and pain behavioral tests. CFA was intraplantarly administrated to induce a chronic pain model. 2 μ g folded with or without capsaicin (200 ng) was injected on day one after CFA injection. **(B)** Effect of folded s-RhTx on CFA-induced heat hyperalgesia. BL, baseline. $n = 6$ mice in

each group. (C) Effect of folded s-RhTx on CFA-induced mechanical allodynia. $n = 6$ mice in each group. $*P < 0.05$ compared to “Cap + saline”, two-way ANOVA followed by Bonferroni’s post hoc test. All data were shown as mean \pm s.e.m.

4 In Figure 6e, folded s-RhTx only without capsaicin should be added as a control to rule out the possibility that s-RhTx itself is toxic and leads to the degeneration of IENF.

Response: We thank the reviewer for the constructive suggestions. As suggested, we included the folded s-RhTx only as a control and performed additional experiments to test whether s-RhTx itself is toxic and leads to the degeneration of IENF. We found that folded s-RhTx only without capsaicin did not induce spontaneous pain (Fig. 6A, B in the revised manuscript and Figure R1) and IENF degeneration (Fig. 6E and Fig. S7 in the revised manuscript). These data suggest that s-RhTx itself is not toxic.

5 Would the degeneration of IENF mediated by s-RhTx and capsaicin lead to local neuroinflammation or recruitment of immune cells? Also, there are several ways of degeneration of IENF (e.g., “dying back” of the axon). What is the type of IENF degeneration in this study?

Response: We thank the reviewer for raising this interesting issue. We conducted experiments to test whether IENF degeneration mediated by s-RhTx and capsaicin could lead to local inflammation or recruitment of immune cells. New result showed that there was no obvious local inflammation in the hind paw injected with folded s-RhTx and capsaicin (Figure R3A). H&E staining also showed that application of s-RhTx and capsaicin did not induce the recruitment of immune cells in the injected paw (Figure R3B) at 1 week after the injection, when the robust degeneration of IENF occurred.

Co-application of s-RhTx with capsaicin could cause calcium overload in TRPV1⁺ IENF and trigger “dying back” of these TRPV1⁺ axon degeneration in a distal-to-proximal manner. However, this IENF degeneration was fully recovered in a few weeks and no DRG neuronal death was induced (Fig. S7A-E). It seems that IENF degeneration in this study is a kind of “dying back” of TRPV1⁺ axon, which is also a transient and reversible process. Future study is warranted to further clarify the detailed mechanism for the degeneration and regeneration of IENF.

Figure R3. Plantar injection of s-RhTx didn't induce inflammatory responses in mice. (A) Statistical results of hind paw thickness in mice. 2 μg folded s-RhTx with capsaicin (200 ng) in 20 μl saline was injected. N.S., not significant, two-way ANOVA. $n = 9$ mice in each group. (B) H&E staining showing the results of cell staining in hind paw skin of mice after 1 week treatment by folded s-RhTx with or without low dose capsaicin. N.S., not significant, $n = 3$ mice in each group. Scale bar, 50 μm . All data were shown as mean \pm s.e.m.

6. local injections of s-RhTx are not likely to change body temperature. If the authors wish to discuss the effect of s-RhTx/capsaicin on body temperature (or other potential side effects), a different experiment should be performed.

Response: We thank the reviewer for the insightful comments. We also agree with the reviewer that the local injections of s-RhTx in this study are not likely to change body temperature. To test the effect of systemic administration of s-RhTx/capsaicin on body temperature, we performed intraperitoneal injection of linear-s-RhTx or folded s-RhTx (10 mg kg^{-1}) with capsaicin (1 mg kg^{-1}) in mice and measured body temperature. The results showed that systemic administration of s-RhTx with capsaicin didn't change body temperature in mice (**Figure R4**).

Figure R4. Systemic administration of s-RhTx with capsaicin didn't alter body temperature in mice. Rectal temperatures in mice were measured before treatment (BL) and at 0.5, 1, 3, 24 and 72 hours following intraperitoneal (*i.p.*) administration of linear-s-RhTx or folded s-RhTx (10 mg kg^{-1}) with capsaicin (1 mg kg^{-1}). $n = 6$ mice in each group. No significance in two-way ANOVA. All data were shown as mean \pm s.e.m.

Reviewers' Comments:

Reviewer #1:

Remarks to the Author:

All previous comments have been addressed by the authors in the revised manuscript and I believe the work is now suitable for publication in Nature Communications.

Reviewer #2:

Remarks to the Author:

The authors provided new data to address the critiques. New results shown in the response letter are important for the paper and should be included in the main paper for publication.